# Mts1 (S100A4) and Its Peptide Demonstrate Cytotoxic Activity in Complex with Tag7 (PGLYRP1) Peptide

**DOI:** 10.3390/ijms25126633

**Published:** 2024-06-16

**Authors:** Daria M. Yurkina, Elena A. Romanova, Kirill A. Shcherbakov, Rustam H. Ziganshin, Denis V. Yashin, Lidia P. Sashchenko

**Affiliations:** 1Institute of Gene Biology (RAS), Moscow 119334, Russia; yrkina121@gmail.com (D.M.Y.); elrom4@rambler.ru (E.A.R.); sashchenko@genebiology.ru (L.P.S.); 2Institute of Biomedical Chemistry, Pogodinskaya Street 10, Moscow 119121, Russia; kirill.soff@gmail.com; 3Shemyakin & Ovchinnikov Institute of Bioorganic, Chemistry, Russian Academy of Sciences, ul. Miklukho-Maklaya, 16/10, Moscow 117997, Russia; ziganshin@mail.ru

**Keywords:** PGLYRP1, S100A4, TNFR1, cytotoxicity, tumor cells, apoptosis, necroptosis, short peptides, thermophoresis, molecular docking

## Abstract

Receptors of cytokines are major regulators of the immune response. In this work, we have discovered two new ligands that can activate the TNFR1 (tumor necrosis factor receptor 1) receptor. Earlier, we found that the peptide of the Tag (PGLYRP1) protein designated 17.1 can interact with the TNFR1 receptor. Here, we have found that the Mts1 (S100A4) protein interacts with this peptide with a high affinity (K_d_ = 1.28 × 10^−8^ M), and that this complex is cytotoxic to cancer cells that have the TNFR1 receptor on their surface. This complex induces both apoptosis and necroptosis in cancer cells with the involvement of mitochondria and lysosomes in cell death signal transduction. Moreover, we have succeeded in locating the Mts1 fragment that is responsible for protein–peptide interaction, which highly specifically interacts with the Tag7 protein (K_d_ = 2.96 nM). The isolated Mts1 peptide M7 also forms a complex with 17.1, and this peptide–peptide complex also induces the TNFR1 receptor-dependent cell death. Molecular docking and molecular dynamics experiments show the amino acids involved in peptide binding and that may be used for peptidomimetics’ development. Thus, two new cytotoxic complexes were created that were able to induce the death of tumor cells via the TNFR1 receptor. These results may be used in therapy for both cancer and autoimmune diseases.

## 1. Introduction

Understanding the interaction mechanisms of cytokines and their analogues with specific receptors is the basis of immunotherapy [1,2]. The binding of cytokines to receptors on the cell surface induces many cellular functions that are often diametrically opposed [3]. A striking example of this is TNFR1 (tumor necrosis factor receptor 1), which, depending on the activation of ubiquitinase, can activate intracellular processes, leading to both the proliferation of immune or tumor cells and their death [4]. In the first case, the NFkB transcription factor is involved, and in the second, either caspases or RIP1 kinase participates in apoptosis or necroptosis development [5,6].

Accordingly, the specific ligand of this TNFR1 receptor is a key cytokine in the development of the immune system and the pro-inflammatory immune response, and it has the opposite effect on cells and the vital activity of the body [7]. By inducing cell death, TNF (tumor necrosis factor) can act as an antitumor agent [8,9]. At the same time, cell death under the action of this cytokine may be one of the causes of the destruction of cartilage and bone tissues during the development of autoimmune arthritis [10]. The expansion of the spectrum of TNFR1 ligands and the understanding of the mechanisms of their action will allow us to outline new approaches for the creation of antitumor and anti-inflammatory therapeutic drugs.

The mechanism of interaction of the TNFR1 receptor with its ligands is widely discussed [11,12,13,14]. TNFR1 has been shown to be able to induce alternative cytotoxic processes in the same cells: apoptosis and necroptosis. After the binding of the ligand to the exodomain of the receptor, the TRADD adapter interacting with the intracellular domain participates in the formation of a protein complex containing also caspase 8 and RIP1 kinase. The activation of caspase 8, followed by the activation of caspase 3, leads to the release of nuclear nuclease, the cleavage of DNA to nucleosomes, and apoptotic cell death. It is believed that active caspase 8 cleaves RIP-1 kinase, blocking necroptosis [15]. In cells where caspase 8 activity is suppressed, blocking apoptosis leads to the activation of RIP1-dependent necroptosis.

We investigated developing cytotoxic processes during TNFR1 activation and found two new proteins interacting with this receptor: the immune system regulator PGLYRP1 (Tag7) and DNA-hydrolyzing autoantibodies. Autoantibodies fully simulated the cytotoxic effect of TNF: they bound to TNFR1 and induced cell death [16]. In this case, Tag7 acts as an inhibitor of the cytotoxic activity of TNF and other TNFR1 ligands. For the manifestation of Tag7 cytotoxic action, the presence of a coactivator, stimulating cytotoxicity, is necessary. During antitumor studies, a cytotoxic complex released by human lymphocytes was identified, and the main heat shock protein Hsp70 was identified as a coactivator of the Tag7 protein [17]. The Tag7- Hsp70 complex, as well as TNF, induces both apoptosis and necroptosis in the cell [18].

Structural and functional analyses of Tag7 made it possible to localize the epitope of this protein necessary for interaction with TNFR1 and Hsp70. Peptide 17.1 (SNYVLKGHRDVQ) corresponds to the C-terminal section of the chain (aa 163–175); moreover, as a full-sized protein, it inhibits the cytotoxic effect of TNFR1 ligands. In combination with Hsp70, it induces the death of tumor cells. It was also possible to separate the activities of this bifunctional peptide. Two fragments were identified and synthesized: a fragment responsible for binding to TNFR1 and inhibiting cytotoxicity (it was designated 17.1A) and a peptide fragment with a high affinity for Hsp70, which is involved in the killing of tumor cells (17.1B) [19].

During a study on the anticancer effect of the Tag7-Hsp70 complex on metastatic cells, the second partner of the Tag7 protein was identified [20]. Tag7 interacts with Mts1 (metastasin 1, S100A4), a member of the S-100 family involved in metastasis processes [21]. Both proteins form the Tag7-Mts1 chemotactic complex. This complex has no cytotoxic activity, and it induces the movement of lymphocytes along the concentration gradient of this complex. Interestingly, Mts1 and Hsp70 compete for binding to Tag7. An excess of Mts1 displaces Hsp70 from the Tag7-Hsp70 complex to form a Tag7-Mts1 complex devoid of cytotoxic activity [22]. These results suggest that both proteins bind to the same site of the Tag7 molecule. It is possible that Mts1, like Hsp70, can bind to peptide 17.1 to form a cytotoxic complex.

The purpose of this work is as follows: (1) to elucidate the ability of peptide 17.1 to interact with Mts1 and form a cytotoxic complex; (2) to characterize the cytotoxic processes induced by this complex; (3) to identify the Mts1 peptide fragment responsible for the formation of the 17.1-Mts1 complex; (4) to establish the binding modes of 17.1 with Mts1 and TNFR1; and to investigate its intermolecular interactions.

## 2. Results

### 2.1. Peptide 17.1 Binds to Mts1

In the previous section, we noted the competition between Hsp70 and Mts1 for the binding center of Tag7. We have shown that Hsp70 interacts with peptide 17.1, which is located at the C-terminal region of the polypeptide chain Tag7. Here, we investigated the interaction of 17.1 with Mts1 using affinity chromatography. The recombinant Mts1 protein (100 ng total protein, 0.1 mg/mL) was applied to CNBr-activated Sepharose with immobilized peptide 17.1. The bound proteins were eluted and analyzed using SDS-PAGE and a subsequent Western blot test with specific anti-Mts1 antibodies. It was shown that Mts1 is present in the eluate in the form of a tetramer (Figure 1). Thus, we can assume that 17.1 binds to Mts.

### 2.2. 17.1-Mts1 Complex Has Cytotoxic Effects on TNFR1-Positive Tumor Cells

Next, we tested the cytotoxic activity of the 17.1-Mts1 complex. Previously, we showed that Tag7-Hsp70 and 17.1-Hsp70 complexes induced cytotoxic activity in mouse cells of the L929 line, human leukemia cells, HL-60, human embryonic kidney cells, HEK293T, and erythroleukemia cells, K562. To identify the identity in the mechanisms of cytotoxic action of the 17.1-Mts1 complex, the same spectrum of target cells was used. The results in Figure 2a indicate that this complex has cytotoxic activity at a 1 nM concentration. In addition to the cytotoxic 17.1-Hsp70 complex, the 17.1-Mts1 complex induces the death of the mouse cells of the L929 line, the human leukemia cells of the HL-60 line, and human embryonal kidney cells of the HEK293T line. Erythroleukemia cells, K562, were resistant to the action of this complex; moreover, this was shown earlier for the Tag-Hsp70 cytotoxic complex. Antibodies to TNFR-1 completely block cytotoxicity. Consequently, this receptor is involved in the transmission of a cytotoxic signal.

As mentioned above, we were able to identify the fragments of peptide 17.1 responsible for binding to TNFR1 and inhibiting the cytotoxic activity of its ligands (17.1A, RSNYVLKG) or for binding to Hsp70 and inducing cytotoxicity (17.1B, HRDVQRT). It can be seen (Figure 2b) that in a complex with peptide 17.1A, Mts1 does not contribute to the appearance of cytotoxicity. However, similarly to the 17.1B-Hsp70 complex, the 17.1B-Mts1 complex exhibits cytotoxic activity. The dependence of cytotoxic activity on the concentration of the 17.1-Mts1 complex was dose-dependent. The maximum value of cytotoxicity is achieved at the 1 nM concentration of the 17.1-Mts1 complex. It should be noted that, unlike the 17.1-Mts1 complex, the complex of this protein with a full-sized Tag7 does not have a cytotoxic effect on cells. To understand the reasons for this difference, a quantitative assessment of the interaction between Mts1 and Tag7 and its peptide fragments was undertaken.

### 2.3. Peptides 17.1 and 17.1B Have a High Affinity for Mts1

To evaluate the protein–protein interaction of Mts1 with Tag7 and with peptides, 17.1, 17.1A, and 17.1B, a quantitative microscale thermophoresis method was used to reliably determine the parameters of the interaction of protein complexes [23,24,25]. Thermophoretic signals obtained by adding peptides and Tag7 to labeled Mts1 demonstrate clear binding curves with an increase in the concentration of studied compounds (Figure 3a–d). The obtained dissociation constants (K_d_) are presented in Table 1.

It can be observed that the interaction of Mts1 with peptides 17.1 and 17.1B is highly specific. The corresponding binding curves show the nanomolar values of the K_d_. It can be assumed that Mts1 forms a strong complex capable of inducing cytotoxicity with these peptides. Mts1 complexes with full-sized Tag7 and peptide 17.1A show a significantly higher apparent K_d_, which indicates a lower affinity of these compounds. Apparently, Tag7 and its peptide fragment 17.1A cannot form a stable complex with Mts1, activating the cytotoxic signal via the TNFR1 receptor.

### 2.4. 17.1-Mts1 Complex Interacts with TNFR1 on the Cell’s Surface

We investigated the ability of peptide 17.1 to form a complex with Mts1 and TNFR1 on the cell surface. The 17.1-Mts1 complex was added to HEK293T cells with TNFR1s on their surface. After incubation, a crosslinking agent (BS^3^) was added, forming covalent bonds between molecules located at a distance of less than 1 nm at the time of addition. Next, the cells were lysed, and membrane proteins were precipitated using anti-TNFR1 antibodies conjugated with magnetic beads. The obtained material was separated using SDS electrophoresis and detected using Western blotting with polyclonal antibodies to Mts1 or Tag7. Figure 4a shows that, in both cases, products with a molecular weight of about 96 kDa are detected, which corresponds to the sum of the masses of TNFR1 (55 kDa), tetrameric Mts1 (40 kDa), and peptide 17.1 (1.5 kDa). These results indicate that 17.1-Mts1 interacts with TNFR1 on the surface of HEK293T cells. It should be noted that recombinant Mts1 alone also binds to the TNFR1 receptor on the cell surface in the experimental conditions mentioned above, forming several products, ranging from 85 to 60 kD (Figure 4b).

### 2.5. 17.1-Mts1 Complex Induces Apoptosis and Necroptosis in Tumor Cells

As was shown above, the TNFR1 receptor induces alternative processes that develop at different time intervals: apoptosis after 3 h and necroptosis after 20 h of TNFR1 ligand interaction with the cell [11]. We have previously shown that the cytotoxic complex 17.1-Hsp70 induces both apoptosis and necroptosis. Here, we have shown that cytotoxic processes in tumor cells activated by the action of 17.1-Hsp70 and 17.1-Mts1 complexes are identical. It can be seen that after 3 h of interaction with the 17.1-Mts1 complex, caspase-dependent apoptosis develops in cells. Cell death was almost completely blocked by caspase inhibitors (Figure 5a). One can also see that the main effector caspase, caspase 3, is activated during first hour of 17.1-Mts1 complex treatment. (Figure 5b) After 20 h, cell death was prevented in the presence of necrostatin1, an inhibitor of RIP1 kinase, which inhibits necroptotic processes (Figure 5c). Downstream necroptotic kinases RIP3 and MLKL are phosphorylated one hour after 17.1-Mts1 addition, supporting necroptosis development. (Figure 5d).

### 2.6. Peptide 17.1 Forms a Cytotoxic Complex with Mts1 Peptide Fragments

Next, we localized the Mts1 peptide site responsible for the formation of the cytotoxic complex. Mts1 was subjected to a limited tryptic hydrolysis to obtain peptides with a molecular weight of 1–2 kDa, which is optimal for analysis and subsequent amino acid synthesis. The gel filtration of four-hour hydrolysis products on the Superdex peptide column revealed several fractions with a molecular weight of 1–3 kDa. To identify the active fragment, peptide 17.1 was incubated with an aliquot of each fraction, and cell death was tested under the action of the resulting mixture of peptides. The results are presented in Table 2. It can be observed that peptide 17.1 exhibited cytotoxic activity only in a mixture with peptides of fraction No. 7. (The first fraction contains traces of nonhydrolyzed Mts1 protein)

A MALDI analysis of this fraction made it possible to establish the amino acid sequence of the major peptide of seventh fraction, designated the M7 (ELPSFLGKRTDEAAFQK) peptide, located in the central part of the Mts1 molecule. (The MALDI Peak table is located in the Appendix A.) This peptide was synthesized, and its cytotoxic activity, in complex with functional Tag7 peptides, was investigated. The results shown in Figure 6a indicate that the M7 peptide can interact with peptides 17.1 and 17.1B, as well as with the full-size Tag7 molecule, to form cytotoxic complexes; these complexes induced cell death through interacting with the TNFR1 receptor. The 17.1-M7 complex had the greatest cytotoxicity, and the cytotoxic activity of the 17.1B-M7 and Tag7-M7 complexes was significantly lower. The 17.1A-M7 complex showed no noticeable cytotoxic effect on tumor cells.

The effect of the concentration of the 17.1-M7 complex on cytotoxic activity was dose-dependent. The maximum cytotoxicity was also achieved at a concentration of 1 nM, as is shown for complex 17.1 with full-size Mts1 (Figure 6b). It can also be seen that the M7 peptide, unlike the full-size Mts1, can interact with the full-size Tag7 to form a cytotoxic complex. A comparative quantitative assessment of the interaction of Tag7 with Mts1 and its peptide using microscale electrophoresis allowed us to establish that the peptide fragment has a significantly higher affinity for Tag7 (K_d_ = 2.96 nM), and these protein components can form a stable Tag7-M7 complex (Figure 6c).

Next, cytotoxic mechanisms induced in tumor cells by the 17.1-M7 complex were investigated. It can be seen that this complex induces caspase-dependent apoptosis and RIP1-dependent necroptosis in tumor cells. After 3 h of incubating the complex with cells, cytotoxicity was blocked by inhibitors of caspases 3 and 8 after 20 h by the inhibitor of RIP1-phosphokinase necrostatin (Figure 7a).

We previously reported that the Tag7-Hsp70-induced complex and FasL necroptotic signals are transmitted with the participation of lysosomes and mitochondria. Here, we tested the role of these key organelles in the development of 17.1-M7-dependent necroptosis (Figure 7b). It can be seen that cytotoxic activity decreases in the presence of EGTA, inhibitors of Ca^2+^-dependent proteases, calpains, and lysosomal cathepsins B and D, as well as the antioxidant ionol. These results suggest that both lysosomes and mitochondria are involved in necroptotic processes.

Thus, the Mts1 peptide fragment was identified, which retains the ability of a full-sized protein to form a cytotoxic complex with Tag7 peptide fragments. Interestingly, this fragment, unlike the full-size Mts1 protein, can shape cytotoxic complexes with full-size Tag7 proteins.

### 2.7. Interactions of 17.1 Peptide with Mts-1 and TNFR1 Revealed by Molecular Modeling

Biochemical data clearly show that the 17.1 peptide interacts with TNFR1 [20]. To obtain the possible binding mode of the peptide with the receptor, docking calculations were also conducted. According to the docking data, 17.1 binds on the surface of TNFR1 in an extended S-like conformation in a corresponding groove (Figure 8a). 

To evaluate the stability of the complex, it was used as a starting structure for a 100 ns MD-run. The root mean square deviation (RMSD) of the backbone atoms of the 17.1 peptide during the simulation is shown in Figure 9. The RMSD reaches high values (up to 12 Å, with a mean value of 7~8 Å), which indicates the considerably conformational changes in the peptide during the simulation.

To investigate the conformational changes occurring for the peptide, the MD trajectory was clustered based on the RMSD values of backbone atoms of 17.1 with a cutoff of 2.0 Å. Fifteen clusters were obtained, with four clusters describing ~90% of conformations. The representative conformations of clusters 1–4 are presented in Figure 8b–e. One can easily see that in clusters 1, 3, and 4, the 17.1 peptide changes its “S-like” conformation to a more “C-like” conformation, with an N-terminus exposed to the solvent. The conformation of the protein in these clusters also changes, the groove that accommodates the docked peptide in the crystal structure disappears, and the surface of the supposed binding region flattens. Despite this, the conformation of the peptide in cluster 2 remains “S-like”, and a large groove forms on the surface of the protein in the binding region.

The interactions that are present between 17.1 and the protein were also calculated for the considered clusters and are presented in Appendix A. The residues of 17.1 that interact with TNFR1 in all representative conformations of every considered cluster are Tyr144, Arg150, Val152, Arg154, and Thr155. On the other hand, the analogous protein residues are Cys56, Ser72, Lys75, Arg77, and Glu79. The dominant interactions established between 17.1 and TNFR1 are highly specific and strong, as they include hydrogen bonds and salt bridges.

According to experimental data described above, the binding region of Mts1 for the 17.1 peptide is a sequence, ^41^ELPSFLGKRTDEAAFQK^57^ (M7 peptide), which forms four identical secondary structures. These structures are located at the four opposite ends of the tetramer and form a “helix”–“unstructured region”–“helix” motif (Appendix A). 

Stoichiometry shows that the binding ratio of 17.1:Mts1 is 2:1, so each Mts1 tetramer should possess two binding sites for 17.1 peptides. The only reliable such sites, in our opinion, are two identical grooves located near the M7 binding sequence of A and C subunits (Appendix A). So, such a groove of the A subunit was chosen as a target for molecular docking. One can easily see that the 17.1B part of the docked 17.1 peptide is located on the surface formed by the M7 sequence (Figure 10a). Meanwhile, the 17.1A portion of 17.1 forms a helix-like structure and cannot be fully accommodated by the protein. We supposed that a groove under the 17.1A of the 17.1 on the protein surface would be widened in its real structure to fully bind the peptide (Figure 10a). 

To test this hypothesis and to evaluate the stability of the obtained Mts1-17.1 complex, a 100 ns MD simulation was run. Figure 9 illustrates the root mean square deviation (RMSD) of the atomic position for the backbone atoms of the 17.1 peptide. The RMSD values reach 11 Å, which indicates that the 17.1 peptide undergoes significant conformational changes.

Then, the full trajectory of the Mts1-17.1 complex was clustered with a cutoff = 1.5 Å. So, 10 clusters were obtained in total, and the first most populated cluster described more than 87% of conformations, so this cluster could be considered the dominant conformation of the 17.1 peptide throughout the whole trajectory. The conformation represented by the first cluster is shown in Figure 10b. It can be seen that the groove that was narrow in the crystal structure significantly enlarges during MD and accommodates the 17.1A N-terminus portion of 17.1, so 17.1 becomes fully bound with the protein.

The interactions formed by the 17.1 peptide in cluster 1 and the protein are summarized in Appendix A and visualized in Appendix A. The peptide is mainly bound with the protein by hydrogen bonds, but some salt bridges and hydrophobic interactions are also present. Five of seven residues from the 17.1B portion of 17.1 interact with residues of the M7 portion of Mts1, forming interactions as strong and specific as H-bonds and salt bridges. According to the interaction analysis, the 17.1A subpeptide plays a significant role in the binding of Mts1, which greatly corresponds with the experimental data. The 17.1 peptide forms two ionic interactions via Asp151 and Arg154 with the Lys57 and Glu91 of the Mts1, respectively.

## 3. Discussion

An interesting result of this work was the production of a new cytotoxic complex consisting of a functional fragment of the Tag7 protein (PGLYRP1) and a full-sized Mts1 protein (S100A4). This complex is capable of causing the cell death of TNFR1-carrying tumor cells. Similarly to the TNF cytokine and Tag7-Hsp70 or 17.1-Hsp70 cytotoxic complexes, the 17.1-Mts1 complex interacts with the TNFR1 receptor and induces alternative cytotoxic processes in tumor cells: apoptosis and necroptosis. The K_d_ 17.1-Mts1 of the complex was 12.9 nM, and it was comparable to the K_d_ 17.1-Hsp70 complex (4 nM), which was obtained by us in a previous study [14]. A low K_d_ indicates a high affinity of the components of this cytotoxic complex and its stability.

As mentioned above, Tag7 can change its functions. By binding to TNFR1, it can prevent the interaction of this receptor with other ligands and act as an inhibitor. In complex with Hsp70, it reverses its function and activates cytotoxic processes in TNFR1-carrying tumor cells. Recently, we were able to identify the peptide fragments of Tag7 that were responsible for each function. Here, we have shown that, similarly to Hsp70, Mts1 has a low affinity for peptide 17.1A, which inhibits the cytotoxic activity of TNFR1 ligands. Only the 17.1B-Mts1 complex had cytotoxic activity. All of the above indicates the functional identity of cytotoxic complexes 17.1-Mts1 and 17.1-Hsp70. Interestingly, the full-sized Tag7 does not form a cytotoxic complex with the Mts1 protein [21]. The K_d_ of the Tag7-Mts1 complex was 9210 nM, which explains the instability of the complex. The K_d_ of the 17.1- Mts1 complex is significantly lower. It may be easier for the shortened Tag7 fragment to bind to the Mts1 epitope, which ensures the interaction of these proteins.

The obtained results expand the understanding of the functional activity of Mts1. The proteins of the S-100 family attract close attention by participating in metastasis processes [26,27,28,29]. We have shown that it can participate in immune and antitumor protection. Mts1, presented on the CD3^+^CD4^+^ T-lymphocyte membrane, promotes the interaction of cytotoxic T-lymphocytes with the tumor cell membrane [21]. The Tag7-Mts1 complex induces the chemotaxis of cytotoxic lymphocytes. Here, we have shown that in the complex with peptide 17.1, Mts1 can activate the TNFR1 receptor and induce the death of tumor cells.

An interesting result of this work was the production of two new cytotoxic complexes: complexes 17.1-Mts1, consisting of a functional peptide of the Tag 7 protein (PGLYRP1) and a full-sized protein Mts1 (S100A4), and complex 17.1-M7, consisting of the Tag7 peptide and a functional fragment of Mts1.

Using limited tryptic hydrolysis, it was possible to identify a section of the polypeptide chain of the Mts1 protein responsible for its cytotoxic activity in complex with peptide 17.1. The synthesized 17-membered peptide (ELPSFLGKRTDEAAFQK) corresponding to this section of the polypeptide chain also formed a cytotoxic complex with peptide 17.1. This peptide modeled the function of a full-sized Mts1. The 17.1-M7 complex also interacted with the TNFR1 receptor and induced alternative processes of programmed cell death: apoptosis and necroptosis. Just like Mts1, the M7 peptide did not interact with the inhibitory peptide fragment 17.1A, but also, like Mts1 and Hsp70, was a coactivator of cytotoxic activity in complex with 17.1B. Unlike the full-size Mts1, its peptide fragment had a high affinity for the full-size Tag7 (K_d_ = 2.96 nM) and could form a cytotoxic complex with this protein. Apparently, the shortened fragment of Mts1 has freer access to the active site of Tag7.

It is known that the TNFR1 receptor induces alternative processes of the program of the cell’s death: apoptosis and necroptosis [30,31,32]. Previously, we showed that necroptosis induced by the Tag7-Hsp70 complex is carried out with the participation of lysosomes and mitochondria. Here, we have shown that these cellular organelles are also involved in the development of 17.1-M7-induced necroptotic processes. Inhibitory analysis revealed that the induction of the necroptotic signal takes place with the participation of Ca^2+^-dependent proteases, calpains, cathepsins B and D, and the antioxidant ionol. Calpains are capable of destabilizing the lysosomal membrane and provide the release of lysosomal enzymes into the cytosol [33,34]. Cathepsins are most stable at a physiological cytoplasmic pH. It is believed that they play a key role in the implementation of cell death [35]. Cathepsins are also able to initiate the permeability of the mitochondrial membrane, which leads to damage to the functioning of the respiratory chain and to the ROS complex [36,37]. Antioxidants block the action of ROS [38,39].

It can be seen that the 17.1-M7 complex, including two shortened peptides, induces the same cytotoxic processes as the 17.1-Mts1 complexes, as well as Tag7-Hsp70 with a molecular weight of 90 kDa. We can assume that we really managed to identify the active fragments of the Tag7 and Mts1 proteins that provide cytotoxic activity. Thus, we have created a new cytotoxic complex that induces the death of tumor cells via the TNFR1 receptor. The results obtained can be used in antitumor therapy.

We also studied interactions of the 17.1 peptide with Mts1 and TNFR1 using molecular docking and molecular dynamics. For Mts1, it is shown that the 17.1 peptide forms a stable complex with it after initial conformational changes. We also identified amino acid residues both for the 17.1 and Mts1, which are responsible for the intermolecular interactions. The complex is formed via multiple hydrogen bonds and ionic interactions, and hydrophobic contacts are also present. We suppose that a found binding site for 17.1 may be considered druggable due to how it is formed by a deep and wide groove. And the presence of such specific interactions as hydrogen and ionic bonds, combined with the knowledge of the interacting residues, makes it possible to develop highly affine and specific low-molecular-weight peptidomimetics that will also possess protective properties like the original 17.1 peptide. As for the 17.1-TNFR1 complex, it was shown that it is less stable, and the 17.1 peptide has a high mobility on the surface of TNFR1. Despite this, we also identified some key amino acids both for 17.1 and TNFR1 that are responsible for the complex formation. These residues also form multiple hydrogen and ionic bonds, which makes the 17.1-TNFR1-interacting interface potentially druggable. So, we conclude that the data obtained from the molecular modeling stage of our work may serve as a base for the development of a new class of drug molecules with antitumor activity.

## 4. Materials and Methods

### 4.1. Cell Cultivation and Sorting

Experiments were performed with the L929 mouse fibroblast cell line, which was cultured, respectively, in DMEM (Himedia Laboratories Private Limited, Maharashtra, India) with 2 mM L-glutamine, 10% FCS (Cytiva Livescience™, Marlborough, MA, USA), and antibiotics (penicillin and streptomycin) (Thermo Fisher Scientific, Waltham, MA, USA) at 37 °C in an atmosphere containing 5% CO_2_. The HEK293T human embryonic kidney cell line was cultured in DMEM/F12, 10% FCS (Cytiva Livescience™, Marlborough, MA, USA), and 1% penicillin-streptomycin (Thermo Fisher Scientific, Waltham, MA, USA) at 37 °C in an atmosphere containing 5% CO_2_. The human erythroblastoma K562 cell line and human leukemia HL-60 cell line were cultured in RPMI-1640 (Himedia Laboratories Private Limited, Maharashtra, India), 10% FCS (Cytiva Livescience™, Marlborough, MA, USA), and 1% kanamycin sulfate (Thermo Fisher Scientific, Waltham, MA, USA). These cell lines were obtained from the cell line collection of the N. N. Blokhin National Medical Research Center of Oncology of the Ministry of Health.

### 4.2. Proteins and Antibodies

Recombinant human Mts1 cDNA was subcloned into pQE-30 and expressed in Escherichia coli M15 (pREP4) (Qiagen, Germantown, MD, USA). Recombinant Tag7 protein cDNA (PGLYRP1, GenbankTM Accession Number AF193843) subcloned into plasmid pQE-30 (Qiagen, Germantown, MD, USA) was expressed in E.coli strain M15 [pREP4] (Qiagen, Germantown, MD, USA). The cells were cultured overnight at 37 °C in V = 20 mL of the LB (yeast extract C = 5 g/L NaCl C = 10 g/L, tryptone C = 10 g/L), containing 0.25% kanamycin sulfate (Thermo Fisher Scientific, Waltham, MA, USA). After this, 5 mL of the medium with cells was transferred into flasks containing 500 mL of LB and was cultured for ~4 h until the opt. density OD600 = 0.4–0.6. After this, IPTG was added to C = 1 mM, after which the cells were grown for another ~3 h to OD600 = 0.7–0.9. Then, this was centrifuged using an SX4750A rotor (Beckman Coulter, Brea, CA, USA) at 4000 rpm for 15 min, and the sediment was washed twice with an STE buffer (sodium chloride-tris-EDTA buffer, 1X (pH 8.0)), and centrifuged for 20 min. The cell sediment was dissolved in lysis buffer (20 mM Na_2_HPO_4_ pH = 8) at a rate of 25 mL per 1 L of the original cell culture. Next, the cells were lysed using ultrasound 10 times for 25 s in ice. Next, the cells were centrifuged at 15,000× *g* for 20 min (Beckman Coulter, Brea, CA, USA). The supernatant was collected and filtered through a 0.22 μm filter (Merck, Darmstadt, Germany). The precipitated protein was purified on a HisTrap FF (Cytiva Livescience™, Marlborough, MA, USA) 5 mL ion exchange column. Protein elution was carried out with a linear gradient from 30 mM imidazole (buffer A (20 mM Na_2_HPO_4_, 0.5 M NaCl, pH = 8) + 6% buffer B (20 mM Na_2_HPO_4_, 0.5 M NaCl, 500 mM, pH = 8)) to 500 mM imidazole (100%B). The resulting protein was more than 98% pure, as demonstrated by polyacrylamide gel electrophoresis. The chromatographic runs were performed on AKTA Purifier System and Unicorn 5.0 Software (GE Healthcare, Chicago, IL, USA). Polyclonal antibodies to Tag7 (PGLYRP1) and Mts1 were obtained from ABclonal (ABclonal, Woburn, MA, USA), and polyclonal antibodies to TNFR1 were obtained from Santa Cruz Biotechnology (Santa Cruz Biotechnology, Santa Cruz, CA, USA).

### 4.3. Peptides

Protein Tag7 and Mts1 were hydrolyzed at 37 °C for 3.5 h at a 1:10 trypsin/protein ratio (*w*/*w*) in 50 mM (NH_3_)HCO_3_ (pH 8.0). The hydrolysate was then separated on a Superdex peptide column. Peptides 17.1 (SNYVLKGHRDVQ), 17.1A (RSNYVLKG), 17.1B (HRDVQRT), and M7 were synthesized on an automated peptide synthesizer in accordance with the Fmoc strategy, and HATU (azabenzotriazole tetramethyluronium hexafluorophosphate) was used as a binding agent. Amino acids were taken in an eight-fold excess; the condensation of each amino acid was carried out for 30 min. C-terminal amino acids were attached to the activated resin in the presence of DIPEA for 2 h. After synthesis, the protected peptidyl polymer was washed with diethyl ether, dried, and treated with a mixture of TFA/DTT/H2O/TIS (150/4/3/0.5 wt. %) (15 mL of the mixture per g of peptidyl polymer) for 2 h. The solution was filtered; the untreated peptide was precipitated with a ten-fold volume of diethyl ether and kept at a temperature of 4 °C for 8 h. The precipitated peptide was centrifuged, washed three times with diethyl ether, and dried under vacuum. The untreated peptide was purified by HPLC on a YMC Actus Triart C18 10u 30 × 150 mm column (YMC Europe GmbH, Dinslaken, Germany) in a gradient of 5–55% acetonitrile and lyophilized. The purity of the peptides was tested by HPLC performed on AKTA Purifier System and Unicorn 5.0 Software (GE Healthcare, Chicago, IL, USA).

### 4.4. Affinity Chromatography, Immunoadsorption and Immunoblotting

A column with CNBr-activated Sepharose 4B (GE Healthcare, Chicago, IL, USA) conjugated with 17.1 was prepared as described in accordance with the manufacture’s protocol. The recombinant Mts1 protein was loaded into a column with 17.1-Sepharose 4B. Then, the material was extensively washed with PBS plus 0.5 M NaCl, followed by PBS, eluted with 0.25 M triethylamine, pH 12. The eluted material was resolved using 12% SDS-PAGE and applied to a nitro-cellulose membrane. Polyclonal rabbit antibodies to Mts1 (Abcam, Cambridge, UK; 1:5000; 1 h) followed by HRP-linked antirabbit antibodies (Abcam, Cambridge, UK; 1:10,000; 1 h) were used for detection. The results were visualized using the ECL Plus kit (GE Healthcare, Chicago, IL, USA) in accordance with the manufacturer’s protocol. The chemiluminescence was recorded using iBright (Thermo Fisher Scientific, Boston, MA, USA). HEK293T cells (108 cells) were incubated with the 17.1-Mts1 complex (C = 1 nM). Then, a crosslinking reagent BS3 (Thermo Fisher Scientific, Boston, MA, USA) was added followed by incubation for 1 h at room temperature. This fraction was lysed in the RIPA buffer (Sigma-Aldrich, St. Louis, MO, USA) with the addition of a cocktail of protease inhibitors, RNase inhibitors, and DNase inhibitors (all from Thermo Fisher Scientific, Boston, MA, USA). After sonication, the lysates were purified using Dynabeads (M-280 sheep anti-rabbit IgG; Dynal Biotech ASA, Oslo, Norway), conjugated with anti-TNFR1 antibodies (Santa Cruz Biotechnology, Santa Cruz, CA, USA), in accordance with the manufacturer’s protocol. This material was resolved into 10% SDS-PAGE, followed by Western blotting using antibodies against Tag7 or Mts1 (ABclonal, Woburn, MA, USA). For measuring the expression of markers of apoptosis and necroptosis using Western blotting, such as caspase 3, caspase 9, RIP3, p-RIP3, p-MLKL, or caspase 9, cells under different experimental conditions were lysed immediately with cytoplasm extraction buffer. Protein concentrations were quantified using a Bradford assay kit (Thermo Fisher Scientific, Carlsbad, CA, USA). Total cell lysates were separated by sodium dodecyl sulfate–polyacrylamide gel electrophoresis and transferred to polyvinylidene difluoride membranes (Amersham Biosciences, Buckinghamshire, UK) for blotting with appropriate primary antibodies against β-actin (mouse A2228 at 1:1000, Sigma-Aldrich, St. Louis, MO, USA), human phospho-RIP3 rabbit (#65746 at 1:2000, Cell Signaling Technology, Danvers, MA, USA), human phospho-MLKL (#91689 at 1:1000, Cell Signaling Technology, Danvers, MA, USA), human RIPK3 (#13526 at 1:1000, Cell Signaling Technology, Danvers, MA, USA), caspase 3 (#14220 at 1:1000), and caspase 9 (mouse #7237 at 1:1000), which were also from Cell Signaling Technology, Danvers, MA, USA. After being washed in TBS, the blots were incubated with ECL™ anti-rabbit IgG, peroxidase-linked species-specific whole-antibody (from donkey) secondary antibody (Cytiva Livescience™, Marlborough, MA, USA) (1:10,000; 1 h), or with ECL™ anti-mouse IgG, peroxidase-linked species-specific whole-antibody (from donkey) secondary antibody (Cytiva Livescience™, Marlborough, MA, USA) (1:10,000; 1 h). The results were visualized using the ECL Plus kit (GE Healthcare, Chicago, IL, USA) in accordance with the manufacturer’s protocol. Chemiluminescence was recorded using iBright (Thermo Fisher Scientific, Boston, MA, USA).

### 4.5. Cytotoxicity Assays

L929, HL-60, and K562 cells were cultured in DMEM (Himedia Laboratories Private Limited, Maharashtra, India) with 2 mM L-glutamine or RPMI-1640 and 10% FCS (Cytiva Livescience™, Marlborough, MA, USA) in a 96-well plate to a density of 4 × 104 cells per well. The medium was then replaced by serum-free media DMEM or RPMI-1640, and the cells were incubated with 17.1-Mts1, 17.1A-Mts1, 17.1B-Mts1, 17.1-M7, Tag7-M7, 17.1A-M7, or 17.1B-M7 at 37 °C, 5% CO_2_. A dead cell count was taken after 3 and 20 h using trypan blue staining and a Cytotox 96 analysis kit (Promega, Madison, WI, USA) in accordance with the manufacturer’s protocol. The caspase 3 activity was determined using the caspase 3 assay kit, Fluorimetric (Sigma-Aldrich, St. Louis, MO, USA). For this, 107 cells were treated with 17.1-Mts1 for 3 h at 37 °C. After this period, 5 μL of the reagent AMC was added to each well, and, after homogenization, samples were incubated for 1 h at room temperature. Then, caspase 3 activity was determined by luminescence (CLARIOstar Plus, BMG LABTECH GmbH, Ortenberg, Germany).

### 4.6. Inhibitor Analysis

The agents used to block the cytotoxic activity were added 1 h prior to cell incubation with the 17.1-M7 (1 nM) complex were as follows: caspase 3 inhibitor Ac-DEVD-CHO (5 μM), calpain (10 μM), caspase 8 inhibitor Ac-IEID-CHO (5 μM), RIP1 kinase inhibitor necrostatin 1 (5 μM), EGTA (2 μM), ionol (1 μM), cathepsin B inhibitor Ca-074Me (10 μM), and cathepsin D inhibitor Pepstatin A (10 μM) (all from Thermo Fisher Scientific, Boston, MA, USA).

### 4.7. Microscale Thermophoresis

To study the dissociation constant, purified Mts1 was fluorescently labeled using Alexa Fluor 633 (Invitrogen, Eugene, OR, USA) in accordance with the manufacturer’s protocol. Mts1 (200 nM) was preincubated for 20 min with Tag7-Mts1, 17.1-Mts1, 17.1A-Mts1, and 17.1B-Mts1 in the dark at room temperature in 16 different concentrations, obtained by sequential dilution. To study the dissociation constant of Tag7 with M7, Tag7 was fluorescently labeled using Alexa Fluor 488 (Invitrogen, Eugene, OR, USA) in accordance with the manufacturer’s protocol. Tag7 (200 nM) was preincubated for 20 min with M7 in the dark at room temperature in 16 different concentrations, obtained by sequential dilution. The samples were loaded into glass capillaries of Monolith NT Capillaries (NanoTemper Technologies GmbH, München, Germany) and analyzed by thermophoresis on a Monolith NT 115 nanotemperature apparatus (10% IR laser power). The signal quality was monitored using a NanoTemper Monolith device to detect possible ligand autofluorescence, deposition, aggregation, or ligand-induced changes in the photobleaching rate. The experiments were carried out in at least three repetitions and processed using affinity analysis software (MO Control v.1.6.1, NanoTemper Technologies GmbH, München, Germany).

### 4.8. MALDI Analysis

The MALDI analysis was performed as described in [40].

### 4.9. Computational Methods

#### 4.9.1. Molecular Docking

Crystal structures of Mts1 (PDB ID: 3NCF) and TNFR1 (PDB ID: 1NCF) were downloaded from Protein DataBank and used as targets for molecular docking. The obtained structures were minimized in the SYBYL 8.1 package using Powell’s method and Gasteiger-Hűckel charges. All non-protein atoms were removed. The 17.1 peptide was taken from a crystal structure of the Tag7 protein (PDB ID: 1YCK) and used as a ligand.

The grid box size was set to 26 × 10 × 10 Å and centered at x = 25.036, y = −26.135, z = 21.139, 18 × 14 × 36 Å, and x = 10.139, y = −3.722, z = 27.306 for Mts1 and TNFR1, respectively.

#### 4.9.2. Molecular Dynamics 

Complexes of the 17.1 peptide with Mts1 and TNFR1 were minimized in SYBYL 8.1 and proceeded for molecular dynamics (MD) simulation in GROMACS 2021. An AMER99SB-ILDN force field was used for all MD calculations. The complexes were solvated with TIP3P water and 0.150 M NaCl. The box size was set to 12 Å from the protein surface. The solvated complexes were minimized for 50,000 steps with the steepest descent method. Two-stage equilibration was used: 2 ns NVT-equilibration followed by 4 ns of NPT-equilibration. During all stages of the equilibration, the movements of protein-heavy atoms were restricted. The equilibrated structures were proceeded for a 100 ns production run. During the production step, a V-rescale thermostat and Parrinello–Rahman barostat were used. The temperature was set to 298 K and pressure for 1 bar. Long-range electrostatics were treated using the Particle Mesh Ewald method (PME). The cutoff for short-range non-bonded interactions was set to 12 Å.

### 4.10. Statistical Analysis

The data are presented as means ± standard deviations. All experiments were repeated at least three times. Data were analyzed using Statistica 6.1 (StatSoft^®^, Tulsa, OK, USA) software using Student’s *t*-tests for experiments on cell treatments with a single agent and a two-way ANOVA for experiments on cell treatments with two or more agents (see individual figure legends). The results are presented as average values ± SDs. The value of *p* < 0.05 was considered statistically significant. GraphPad Prism 6 software was used for data presentation.

## 5. Conclusions

In this study, we have created two new cytotoxic complexes, 17.1-Mts1 and 17.1-M7, that induce the death of tumor cells via the TNFR1 receptor. It appears that several proteins (Mts1 and Hsp70) that can closely interact with the fragments of the PGLYRP1 (Tag7) protein can also activate the TNFR1 receptor in complex with these peptides and induce cell death. Molecular docking experiments show the amino acids involved in peptides’ binding and may be used for peptidomimetics’ creation. The results obtained can be used in antitumor therapies.

## Figures and Tables

**Figure 1 ijms-25-06633-f001:**
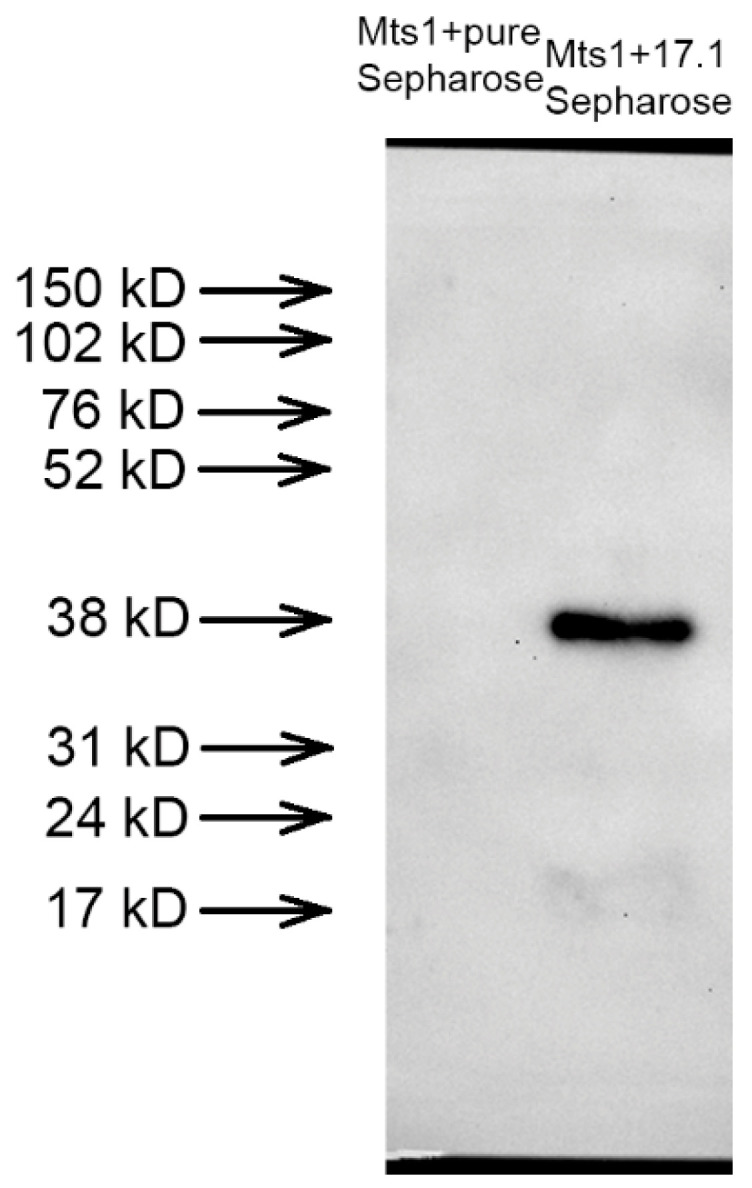
The recombinant Mts1 protein (100 ng) was applied to pure Sepharose (**left probe**) or peptide 17.1 conjugated with Sepharose (**right probe**), eluted with 0.25 M triethylamine, and resolved using SDS PAGE and Western blot tests with antibodies relative to Mts1.

**Figure 2 ijms-25-06633-f002:**
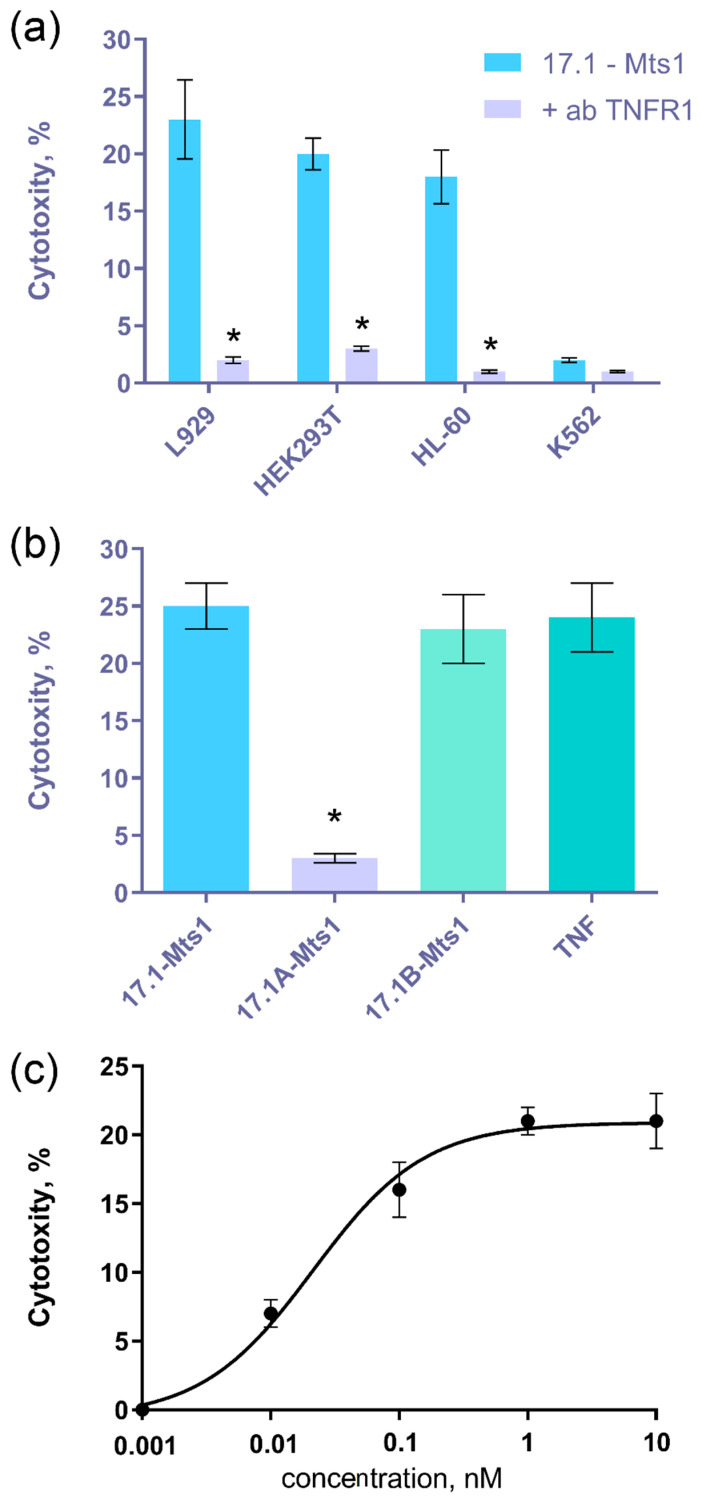
(**a**) Cytotoxic activity of the 17.1-Mts1 complex (1 nM) on L929, K562, HEK 293T, and HL-60 cells in the presence of an antibody to TNFR1 (1:100, 24 h). (**b**) Cytotoxic activity of Mts1 complexes with peptides 17.1, 17.1A, and 17.1B on L929 cells (24 h). (**c**) Concentration dependence of the cytotoxic activity of the 17.1-Mts1 complex on L929 cells (24 h). *n* = 5 for each point (* *p*-value < 0.05).

**Figure 3 ijms-25-06633-f003:**
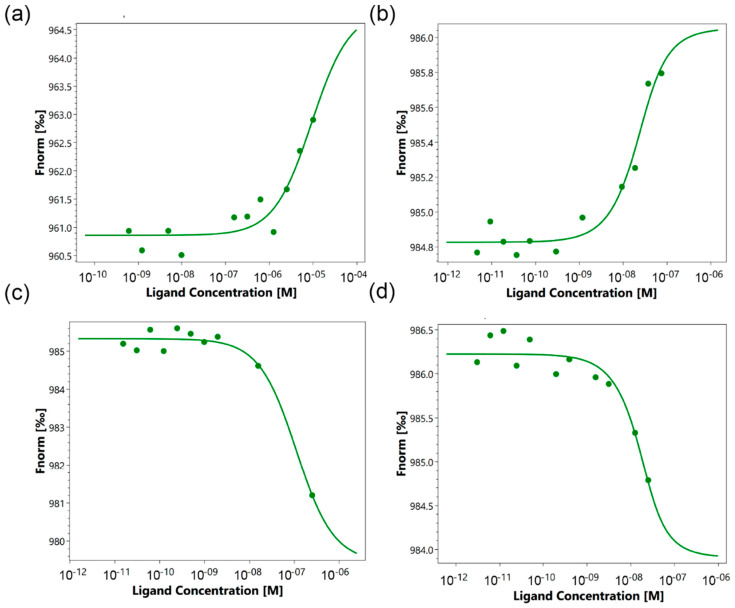
Typical microscale thermophoresis data for the interaction of Mts1 with Tag7 (**a**) and peptides 17.1 (**b**), 17.1A (**c**), and 17.1B (**d**). Each experiment was carried out in triplicate, and the most common data are shown.

**Figure 4 ijms-25-06633-f004:**
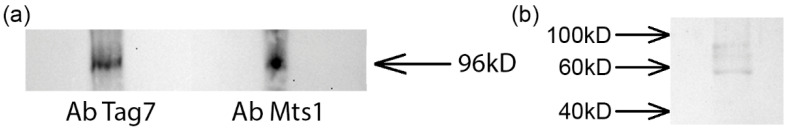
(**a**) The 17.1-Mts1 complex was added to the HEK293T cells and cross-linked with BS^3^ with surface proteins; then, the cells were lysed, the cross-linked material was purified on magnetic beads with anti-TNFR1 antibodies, and after SDS PAGE and Western blotting, the material was stained with antibodies to Tag7 (**left**) or Mts1 (**left**). (**b**) Recombinant Mts1 was added to HEK293T cells and cross-linked with BS3 with surface proteins; then, the cells were lysed, the cross-linked material was purified on magnetic beads with anti-TNFR1 antibodies, and after SDS PAGE and Western blotting, the material was stained with antibodies to Mts1.

**Figure 5 ijms-25-06633-f005:**
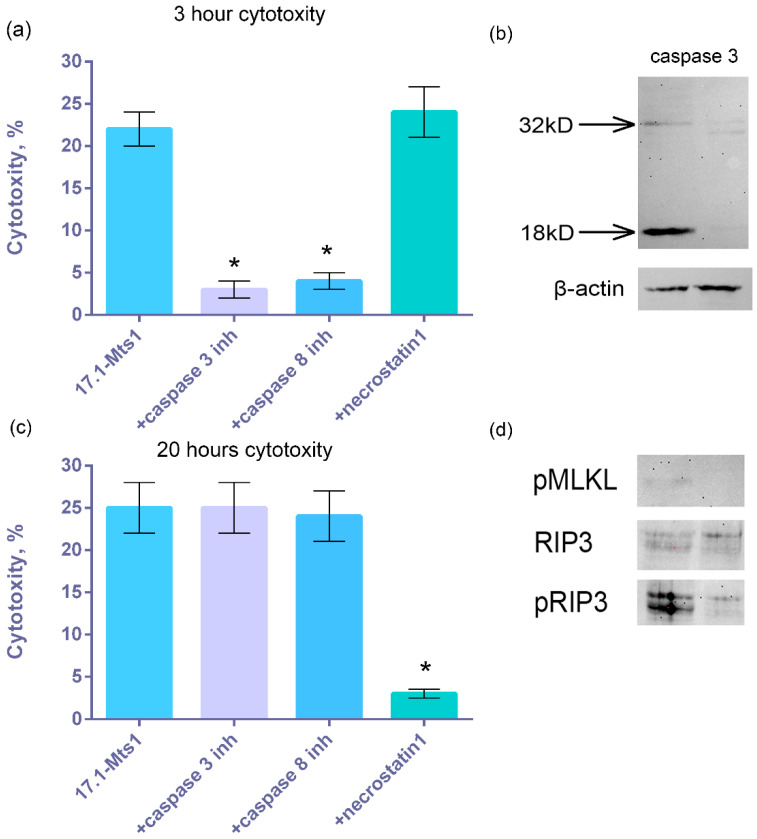
Cytotoxic activity of the 17.1-Mts1 complex on L929 cells after 3 (**a**) and 24 (**c**) hours in the presence of caspase 3 and 8 inhibitors and necrostatin1. *n* = 5 for each point, (* *p*-value < 0.05). (**b**) Western blot with antibodies to caspase 3 from cell lysates 1 h after 17.1-Mts1 addition (left) and control cells (right). (**d**) Western blot with antibodies to phospho-MLKL, RIP3, and phospho-RIP3 from cell lysates 1 h after addition of 17.1-Mts1 (**left**) and control cells (**right**).

**Figure 6 ijms-25-06633-f006:**
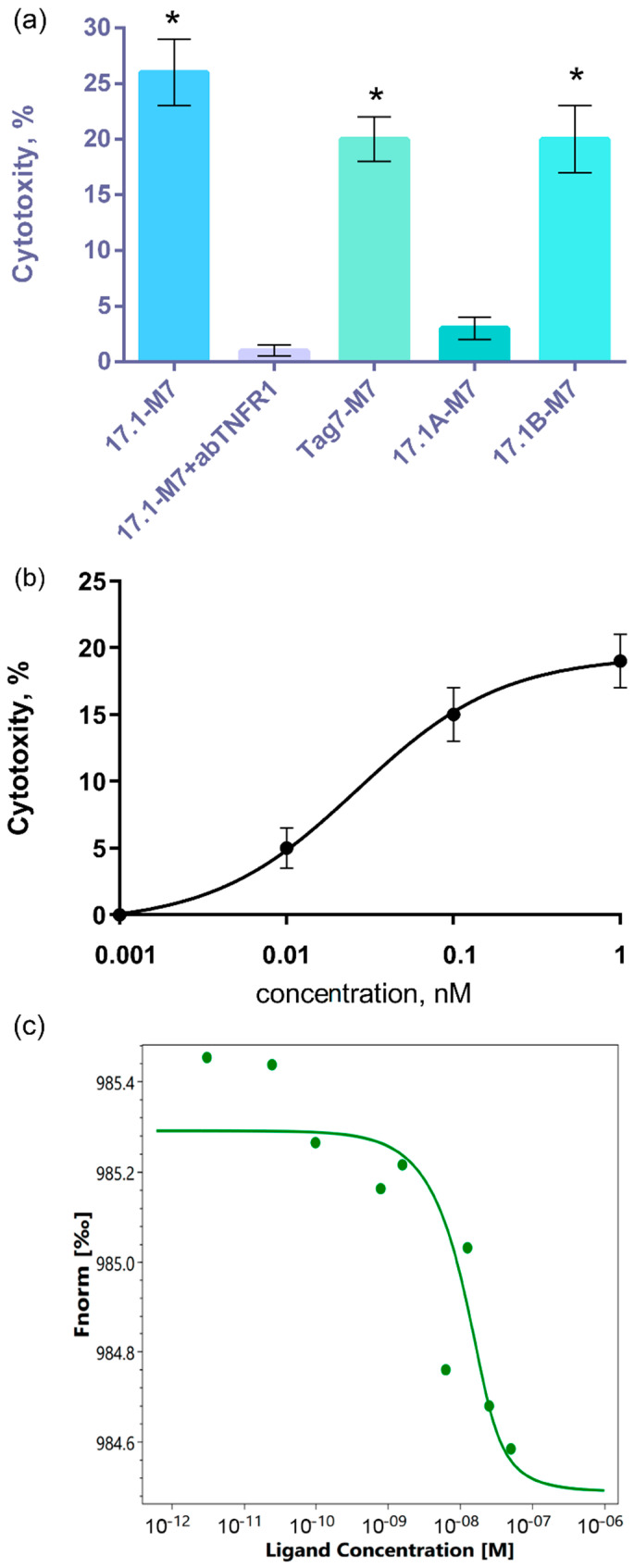
(**a**) Cytotoxic activity of a mixture of peptide M7 with full-size Tag7 and its peptides after 24 h in the presence of anti-TNFR1 antibodies. (**b**) Concentration dependence of the cytotoxic activity of mixture 17.1 with M7 peptides. (24 h). *n* = 5 for each point (* *p*-value < 0.05). (**c**) Microscale thermophoresis data for the interaction of M7 with Tag7. The experiments were carried out in triplicate, and the most common data are shown.

**Figure 7 ijms-25-06633-f007:**
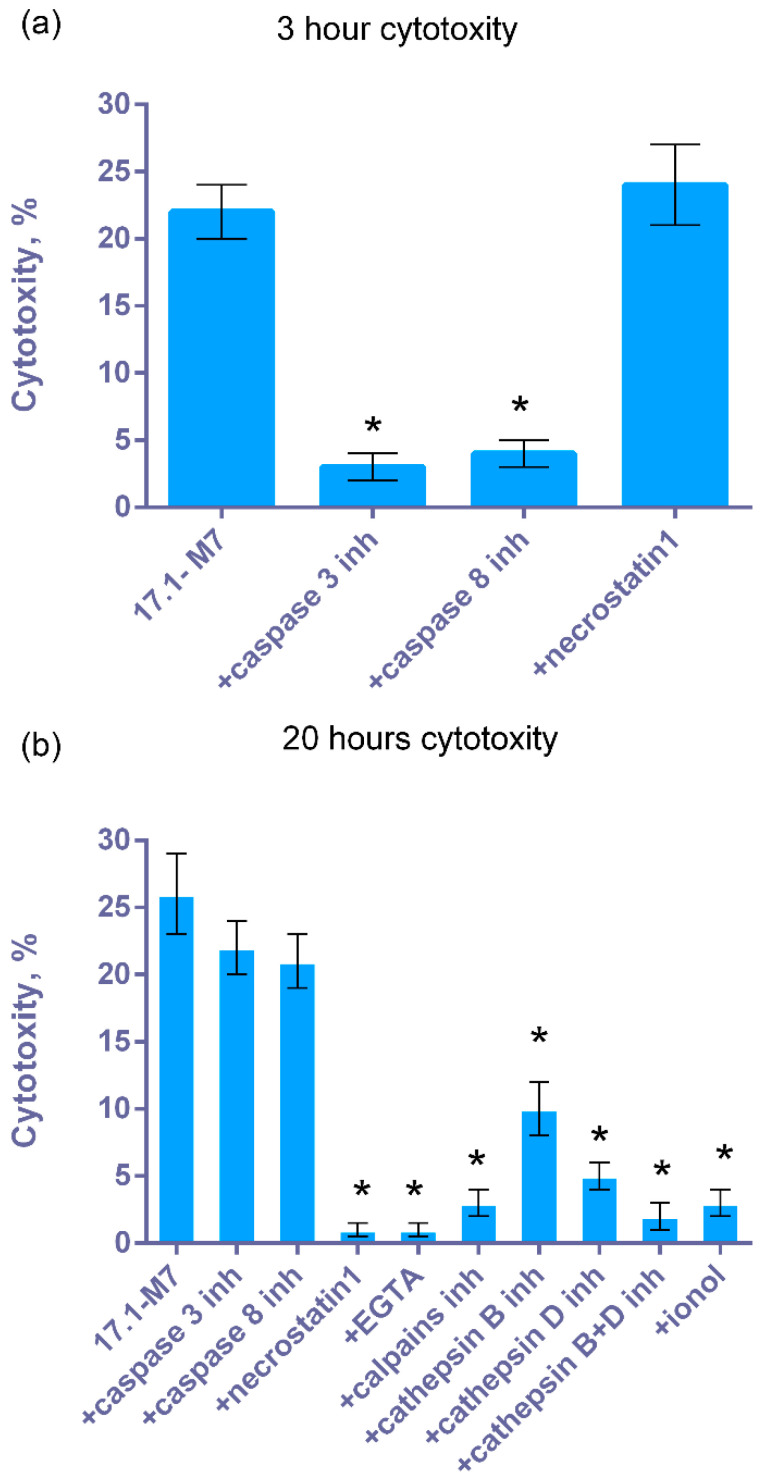
Cytotoxic activity of the 17.1-M7 complex on L929 cells after 3 (**a**) and 20 (**b**) hours in the presence of caspase 3 and 8 inhibitors, necrostatin1, EGTA, calpains and cathepsins inhibitors, and antioxidant ionol. *n* = 5 for each point, (* *p*-value < 0.05).

**Figure 8 ijms-25-06633-f008:**
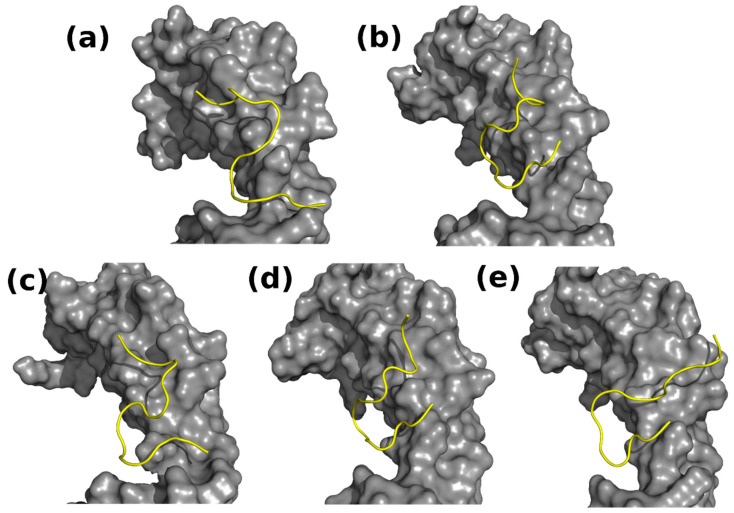
Complexes of TNFR1 and the 17.1 peptide. The protein is shown as a surface and colored in gray; the 17.1 peptide is shown as a cartoon in yellow. (**a**) TNFR1-17.1 complex after docking; (**b**–**e**) TNFR1-17.1 complexes from representative conformations of clusters 1–4, respectively. The TNFR1 crystal structure (**a**) has a narrow groove on its surface, which accommodates the docked peptide in an “S-like” conformation. A similar conformation is adopted by the peptide in cluster 2 (**c**). In the clusters 1, 3, and 4 (**b**,**d**,**e**) the peptide adopts a C-like conformation; the binding surface of the protein flattens and the groove from the crystal structure disappears.

**Figure 9 ijms-25-06633-f009:**
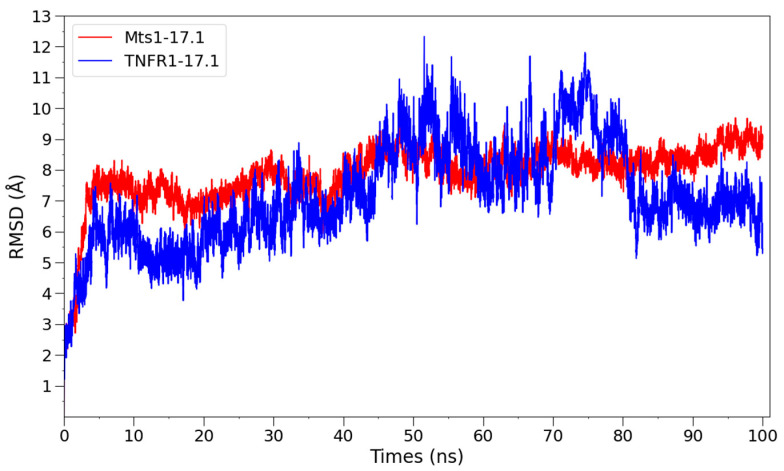
RMSD of backbone atoms of 17.1 peptide in complex with TNFR1 and Mts1. 17.1 peptide in complex with TNFR1 and Mts1 demonstrates high RMSD values. It may indicate that peptides in both complexes go through significant conformational changes in the first ~5 ns of the simulation.

**Figure 10 ijms-25-06633-f010:**
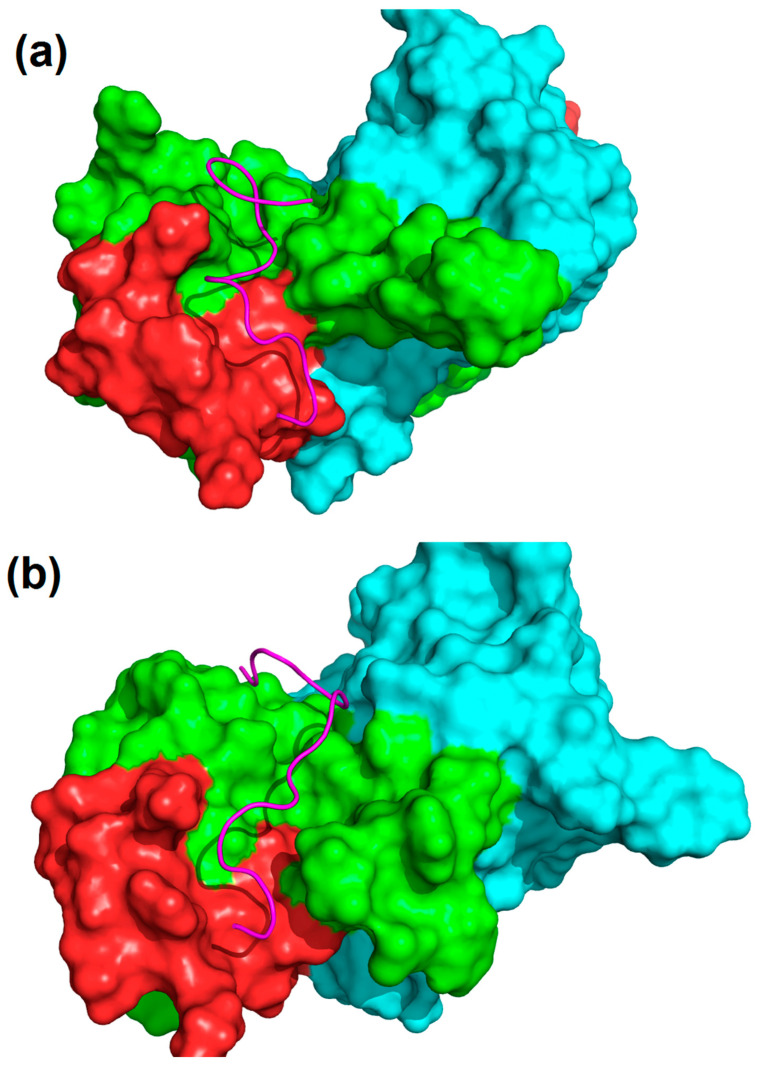
Complexes of Mts1 protein and 17.1 peptide. The protein is represented by the surface, and the peptide is represented by cartoon colored pink. Subunit A is green, and subunit B is blue. Red stands for the portion of Mts corresponding to the M7 peptide, which is responsible for binding with 17.1. (**a**) Mts1-17.1 complex after docking. One can easily see that the N-terminal part of the 17.1 peptide is not bound to the protein and forms a helix-like structure, which seems to be unrealistic. (**b**) Mts1-17.1 complex from the representative cluster of MD. The protein makes a large groove, which accommodates the whole peptide. The peptide now is in an extended conformation and fully interacts with the protein.

**Table 1 ijms-25-06633-t001:** Dissociation constants obtained via microscale thermophoresis.

Ligands	K_d_, nM
Tag7-Mts1	9210 ± 1500
17.1-Mts1	12.9 ± 3
17.1A-Mts1	109 ± 35
17.1B-Mts1	7.26 ± 0.9

**Table 2 ijms-25-06633-t002:** Cytotoxic activity of a mixture of peptide 17.1 with Mts1 hydrolysis fractions separated by gel filtration on a Superdex peptide column.

Fraction Number	Cytotoxicity, %	Fraction Number	Cytotoxicity, %	Fraction Number	Cytotoxicity, %
1	17 ± 3	7	21 ± 4	13	1 ± 0.2
2	6 ± 1	8	3 ± 0.4	14	1 ± 0.3
3	1 ± 0.1	9	1 ± 0.1	15	1 ± 0.4
4	1 ± 0.1	10	1 ± 0.1	16	2 ± 0.2
5	1 ± 0.1	11	1 ± 0.3	17	1 ± 0.3
6	5 ± 1	12	1 ± 0.2	18	5 ± 1

## Data Availability

Data are contained within the article and Appendix A.

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
