# Peer review of "Mts1 (S100A4) and Its Peptide Demonstrate Cytotoxic Activity in Complex with Tag7 (PGLYRP1) Peptide"

_ijms, 2024, doi:10.3390/ijms25126633_

Round 1

Reviewer 1 Report (New Reviewer)

Comments and Suggestions for Authors

In this manuscript the authors found Mts1 and its peptide fragment M7 can form complex with Tag7 protein and exhibited cytotoxicity to cancer cells with TNFR1 receptor on cell surface. Cytotoxicity, binding affinity with Tag 7, apoptosis, necrosis of Mts 1 and its fragments were evaluated in different cancer cell lines. Molecular docking and molecular dynamics results showed complex conformation, hydrogen bonds, salt bridges and hydrophobic interaction between amino acid residues. However, there are some concerns that need to be addressed.

In cytotoxicity studies, L929, HEK293T and HL-60 are sensitive to 17.1-Mts1 complex, however K562 are not response to it, please explain this phenomenon.

Please provide the MALDI-MS spectrum of Mts1 fragments as evidence.

Please provide more detailed or zoom in structure with hydrogen bond and salt bridge interaction in Figure 10.

There are quite a few of typos in the manuscript. Such as “12,9 nm” on line 227, please use decimal point instead of comma. “CD3+CD4+” on line 355, please revise “+” as superscript. “CO_2” on line 418, please correct “2” as subscript. Please correct all of the typos in the manuscript.

Author Response

In this manuscript the authors found Mts1 and its peptide fragment M7 can form complex with Tag7 protein and exhibited cytotoxicity to cancer cells with TNFR1 receptor on cell surface. Cytotoxicity, binding affinity with Tag 7, apoptosis, necrosis of Mts 1 and its fragments were evaluated in different cancer cell lines. Molecular docking and molecular dynamics results showed complex conformation, hydrogen bonds, salt bridges and hydrophobic interaction between amino acid residues. However, there are some concerns that need to be addressed.

In cytotoxicity studies, L929, HEK293T and HL-60 are sensitive to 17.1-Mts1 complex, however K562 are not response to it, please explain this phenomenon.

We thank the reviewer for careful reading of our work and useful comments. In this investigation we have found that 17.1-Mts1 complex executes its activity via TNFR1 receptor. L929 cells, HEK293T and HL-60 cells carry functional TNFR1 receptor and due to this are susceptible to the action of 17.1-Mts1 complex. K562 cells are cells of erythroid origin and do not have functional TNFR1 receptor. That is why they are resistant to the 17.1-Mts1 function.

Please provide the MALDI-MS spectrum of Mts1 fragments as evidence.

In response to this comment we have added peak mass table in the Supplemental material.

Please provide more detailed or zoom in structure with hydrogen bond and salt bridge interaction in Figure 10.

We believe that adding an interaction scheme directly in Figure 10 will make the figure more difficult to perceive. Instead, we have added the figure with h-bond/salt bridge interactions to Figure S3 in the Supplementary Materials.

There are quite a few of typos in the manuscript. Such as “12,9 nm” on line 227, please use decimal point instead of comma. “CD3+CD4+” on line 355, please revise “+” as superscript. “CO_2” on line 418, please correct “2” as subscript. Please correct all of the typos in the manuscript.

We thank the reviewer for useful comments. We have corrected typos and mistakes we have found in the manuscript.

Reviewer 2 Report (New Reviewer)

Comments and Suggestions for Authors

In the manuscript "Mts1(S100A4) and its peptide acquire cytotoxic activity in complex with Tag7 (PGLYRP1) peptide", a team of investigators led by Yurkina, and Sashchenko, describe two novel ligands that can activate TNFR1 receptor. Moreover, the author displays the protein interaction between Tag and Mts1 and further explores the function of this complex. 

Although there were several interesting points made here, this manuscript suffers from several significant weaknesses that may preclude acceptance for publication.

1. In lines 68-75, it is unclear whether the two fragments referred to are peptide 17.1a and 17.1b. Since subsequent results mention 17.1a and 17.1b, more information or background about these two fragments is required in this paragraph.

2. References are needed for the Tag7-Mts1 complex in lines 77-79, such as PMID: 33959389 and PMID: 36303048.

3. More details are needed on how the author identified and found the two new proteins in line 59.

4. Was Figure 1 performed in vitro? Were the Mts1 samples (supposed protein) isolated from the cell line? How much concentration of the Mts1 samples was applied to the Sepharose with peptide 17.1?

5. Multiple functional assays are needed to show the effect of the 17.1-Mts1 complex. These could include proliferation assays, apoptosis assays, and measuring downstream gene expression using western blots, such as caspase 8 and caspase 3.

6. What would happen if Mts1 expression was knocked out? Does the function of peptide 17.1 depend on Mts1? The author needs to perform a knockout assay to evaluate the importance of the 17.1-Mts1 complex.

7. The author needs to perform Co-IP for the interaction between Mts1 and TNFR1 in result 2.4.

8. More functional assays and downstream gene expression data need to be provided in result 2.5. For example, apoptosis and proliferation assays, and measuring the expression of markers of apoptosis and necroptosis using western blot or RT-PCR, such as caspase 3, caspase 9, RIP, p-RIP, MLKL, p-MLKL, RIP3, and p-RIP3.

9. Language editing service is very much needed for this manuscript. some sentences are easy to confuse, such as Line 157.

Author Response

In the manuscript "Mts1(S100A4) and its peptide acquire cytotoxic activity in complex with Tag7 (PGLYRP1) peptide", a team of investigators led by Yurkina, and Sashchenko, describe two novel ligands that can activate TNFR1 receptor. Moreover, the author displays the protein interaction between Tag and Mts1 and further explores the function of this complex. 

Although there were several interesting points made here, this manuscript suffers from several significant weaknesses that may preclude acceptance for publication.

  1. In lines 68-75, it is unclear whether the two fragments referred to are peptide 17.1a and 17.1b. Since subsequent results mention 17.1a and 17.1b, more information or background about these two fragments is required in this paragraph.

We thank the reviewer for careful reading of our work and useful comments. We have added new information in the Introduction Section to provide more information about 17.1A and 17.1B fragments of 17.1 peptide and their functionality.

  1. References are needed for the Tag7-Mts1 complex in lines 77-79, such as PMID: 33959389 and PMID: 36303048.

We thank the reviewer for attention to our work. In response to this comment, we have added new references to the manuscript.

  1. More details are needed on how the author identified and found the two new proteins in line 59.

We have added new information about functionality of Tag7-Hsp70 and Tag7-Mts1 complexes and the history of their activity discovery in the Introduction section.

  1. Was Figure 1 performed in vitro? Were the Mts1 samples (supposed protein) isolated from the cell line? How much concentration of the Mts1 samples was applied to the Sepharose with peptide 17.1?

We apologizes for unclear presentation of our data. The recombinant Mts1 protein (100ng total protein, 0.1 mg//ml) was used for acquisition of the data presented on the Figure 1. We have changed Results Section, Figure 1 legend and Matherials and Methods Sections to correct this issue.

  1. Multiple functional assays are needed to show the effect of the 17.1-Mts1 complex. These could include proliferation assays, apoptosis assays, and measuring downstream gene expression using western blots, such as caspase 8 and caspase 3.

In response to this comment, we have succeeded in purchase in strict time frames the caspase 3 assay and have provided caspase activity data under the effect of the 17.1-Mts1 complex in the manuscript. Additionally the brand new antibodies were purchased and we have added caspase and necroptosis ferments detection experiments using Western blot analysis after the 17.1-Mts1 complex treatment of HEK293T cells.

  1. What would happen if Mts1 expression was knocked out? Does the function of peptide 17.1 depend on Mts1? The author needs to perform a knockout assay to evaluate the importance of the 17.1-Mts1 complex.

We apologize for not presenting our data clearly. In our case Mts1 protein is added in complex with 17.1 peptide to the cells externally and binds to the receptor TNFR1 on the cell surface, it is not originating from any cellular compartments of studied cells. Although some cells may contain Mts1 protein (CSML-100, HeLa) we believe, that internal Mts1 protein do not play any role in the investigated 17.1-Mts1 activity. Mts1 knockout experiment was described in (Metastasis-inducing S100A4 and RANTES cooperate in promoting tumor progression in mice. Forst B, Hansen MT, Klingelhöfer J, Møller HD, Nielsen GH, Grum-Schwensen B, Ambartsumian N, Lukanidin E, Grigorian M. PLoS One. 2010 Apr 28;5(4):e10374. doi: 10.1371/journal.pone.0010374. PMID: 20442771). It does not change the background cell death, but decrease the cancer cell malignity via interference with metastasis process.

  1. The author needs to perform Co-IP for the interaction between Mts1 and TNFR1 in result 2.4.

We have performed additional experiment and presented its data in new Figure 4B.

  1. More functional assays and downstream gene expression data need to be provided in result 2.5. For example, apoptosis and proliferation assays, and measuring the expression of markers of apoptosis and necroptosis using western blot or RT-PCR, such as caspase 3, caspase 9, RIP, p-RIP, MLKL, p-MLKL, RIP3, and p-RIP3.

We have added new functional data, including capase 3 activity assay, measuring via Western blot appearance of cleaved functional caspase 3 (caspase 9 was not responsible for apoptosis signal transduction and presented in Supplemental data) and appearance of phosphorylated forms of necroptosis ferments RIP3 and MLKL short time after addition of 17.1-Mts1 complex to the HEK293T cells. These data were added to the Results, Figure 5 and Matherial and Methods Sections of the manuscript.

  1. Language editing service is very much needed for this manuscript. some sentences are easy to confuse, such as Line 157.

We have tried to improve the language of the manuscript to make it clearer for the potential readers.

Round 2

Reviewer 1 Report (New Reviewer)

Comments and Suggestions for Authors

The revised version addressed my concerns.

Reviewer 2 Report (New Reviewer)

Comments and Suggestions for Authors

The authors' responses in the revised manuscript mostly addressed those critiques adequately. The resulting manuscript is much improved.  It is now suitable for publication in its current form.

This manuscript is a resubmission of an earlier submission. The following is a list of the peer review reports and author responses from that submission.

Round 1

Reviewer 1 Report

Comments and Suggestions for Authors

The manuscript presents a comprehensive study on the interaction between the twelve-membered peptide Tag7 and Mt1, and its subsequent effect on cell death mediated by the TNFR1 receptor. The research topic is timely and of significant importance, shedding light on potential therapeutic avenues. However, there are several areas that could benefit from further refinement.

Introduction:

Broaden the context: The introduction starts with the identification of new ligands for proinflammatory receptors. To provide a more comprehensive background, begin with the significance of understanding immune response mechanisms, especially in the context of tumor cell death. Then, delve into the specifics of proinflammatory receptors.

Clarify the role of TNFR1: The introduction frequently mentions TNFR1 and its diverse roles. For readers unfamiliar with TNFR1, a brief description of its primary function and significance in immune responses would be beneficial.

Methods:

Statistical Analysis (page 9, line 274): Provide more details about the groups being compared. Specify if it's a two-sample t-test and whether it's paired or not.

Peptides (page 8, line 234): While the synthesis of peptides 17.1, 17.1A, and 17.1B is referenced, a brief description of the synthesis method or any unique characteristics of these peptides would enhance clarity.

Figures and Tables:

Figure 1: Improve the resolution. Ensure that the x-labels "1" and "2" are in the same font and aligned on the same horizontal line. Increase the size of the gel image for better visibility.

Figure 2(a): Ensure consistency in legend colors. The legend name for HL-60 is in black, while others are in grey. All should be in black.

Figure 3-D: Adjust the plot size for better visibility of x-labels. The label "1E-06" is not clearly presented. Also, consider resizing the labels A, B, C, and D to be consistent with Figure 2.

Figure 5: Reduce the size of the legends for better proportionality.

Table 2: Resize the table to fit within the page margins. Add explanations about the table's format, such as clarifying the representation of "," between the values "0,2" in the first row of "cytotoxicity %."

Comments on the Quality of English Language

The manuscript is generally well-written, with the English language being clear and understandable. However, there are areas where the phrasing could benefit from improvements for clarity and fluency.

Abbreviations such as "TNFR1", "Mts1", and "Tag7" are used throughout the manuscript. It's crucial to ensure that these are clearly defined upon their first mention and consistently used thereafter. For instance, on page 3, line 88, the term "TNFR-1" is used, while in other places, "TNFR1" is used. Please ensure consistency in terminology.

Rephrased Sentence for Clarity:

Original: (page 2, line 64) "The purpose of this work is as follows: 1) elucidation of the ability of peptide 17.1 to interact with Mts1 and form a cytotoxic complex; 2) characterization of cytotoxic processes induced by this complex; 3) identification of the Mts1 peptide fragment responsible for the formation of the 17.1-Mts1 complex."

Suggested: "This study aims to: 1) Determine how peptide 17.1 interacts with Mts1 to form a cytotoxic complex; 2) Characterize the cytotoxic processes initiated by this complex; and 3) Identify the Mts1 peptide fragment contributing to the formation of the 17.1-Mts1 complex."

Reviewer 2 Report

Comments and Suggestions for Authors

In the current manuscript the authors have investigated the potential of a functional fragment of the protein PGLYRP1 to form a complex with Mts1 protein which induces the death of tumor cells via the TNFR1 receptor. The authors have attempted to prove that the formed complex showed highest cytotoxicity at 1nM concentration. I will recommend accepting the article with some changes.

Provide the significance of the study. Mention the limitations and previous reported studies and the need for further study.

Figure 1 & 4, western blot chromatograph image is not clear. One cannot verify the molecular weight range of the spot mentioned. Please reproduce the results.

Line 167, 168 & 169, “It can be observed that peptide 17.1 exhibited cytotoxic activity only in a mixture with peptides of fraction No. 7. MALDI analyses of this fraction revealed several peptides…..” Please explain the MALDI analysis. Provide the MALDI results if performed.

Authors should elaborate line 224 in a few words. What’s the pharmaceutical importance of the study. How it will be helpful in actual antitumor therapy! Mention representative peptide structure. Authors can also show the small graphical of the obtained study results in the main text. It will be helpful to the readers.

Comments on the Quality of English Language

Minor editing of English language required.